# Retinitis Pigmentosa Masquerades: Case Series and Review of the Literature

**DOI:** 10.3390/jcm12175620

**Published:** 2023-08-28

**Authors:** Abinaya Thenappan, Arjun Nanda, Chang Sup Lee, Sun Young Lee

**Affiliations:** 1USC Roski Eye Institute, Department of Ophthalmology, Keck School of Medicine, University of Southern California, Los Angeles, CA 90033, USA; 2College of Medicine, University of Oklahoma Health Science Center, Oklahoma City, OK 73104, USA; 3USC Ginsburg Institute for Biomedical Therapeutics and Department of Physiology and Neuroscience, Keck School of Medicine, University of Southern California, Los Angeles, CA 90033, USA; 4Department of Ophthalmology, Dean McGee Eye Institute, University of Oklahoma Health Science Center, Oklahoma City, OK 73104, USA; 5Department of Physiology, University of Oklahoma Health Science Center, Oklahoma City, OK 73104, USA

**Keywords:** pseudo-retinitis pigmentosa, retinitis pigmentosa masquerades, vitamin A deficiency, hydroxychloroquine retinopathy

## Abstract

Retinitis pigmentosa (RP) displays a broad range of phenotypic variations, often overlapping with acquired retinal diseases. Timely recognition and differentiation of RP masquerades is paramount due to the treatable nature of many such conditions. This review seeks to present examples of pseudo-RP cases and provide a comprehensive overview of RP masquerades. We first present two pseudo-RP cases, including comprehensive clinical histories and multimodal retinal imaging, to highlight the important role of accurate diagnoses that subsequently steered effective intervention. Subsequently, we conduct an in-depth review of RP masquerades to provide valuable insights into their key distinguishing features and management considerations. The recent approval of ocular gene therapy and the development of investigational gene-based treatments have brought genetic testing to the forefront for RP patients. However, it is important to note that genetic testing currently lacks utility as a screening tool for inherited retinal diseases (IRDs), including RP. The integrity of a precise clinical assessment remains indispensable for the diagnosis of both RP and RP masquerade conditions, thereby facilitating prompt intervention and appropriate management strategies.

## 1. Introduction

The clinical condition known as retinitis pigmentosa (RP) was originally described in 1853, though it wasn’t officially designated as such until 1857 [1,2]. The nomenclature “retinitis pigmentosa” is derived from the Latin terms “retina” and “pigmentum” as well as the Greek word “πυρέττω” (pronounced “pyréttō”), meaning “to inflame”. The term “retinitis” initially suggested an inflammatory nature, which was presumed to underlie the disease. However, contemporary understanding indicates that inflammation is not the primary cause of retinitis pigmentosa. On the other hand, “pigmentosa” refers to the characteristic accumulation of pigment in the retina due to the degeneration of photoreceptor cells, which release their pigment into the surrounding tissue as they deteriorate. Therefore, retinitis pigmentosa (RP), considered by most to be a misnomer, is a term that describes the combination of the two prominent clinical features of the grossly observed retina in RP: retinal degeneration and pigment accumulation. 

In reality, RP represents a cohort of progressive inherited retinal diseases (IRD) characterized by sequential degeneration of rod photoreceptors followed by cone photoreceptors. The initial symptom is reduced night vision, which is followed by a progressive loss of the visual field in a concentric pattern. Fundus abnormalities, often affecting both eyes symmetrically, vary from near normal in early stages to a waxy pallor of the optic nerve head and attenuation of retinal vessels with or without bone spicule pigmentation in the periphery and/or midperiphery in advanced stages [3,4]. 

Given the extensive range of genotypic and phenotypic diversity in RP, coupled with its gradual progression, several inherited retinal diseases (IRD) and non-IRD (acquired) conditions can mimic RP. This complexity can introduce challenges in the diagnostic process and potentially result in delays in appropriate treatment. An accurate diagnosis is paramount due to the distinct therapeutic implications; unlike RP, many of the RP masquerade conditions are treatable and/or reversible when addressed in a timely fashion. While advances in molecular analysis serve as valuable tools for enhancing diagnostic clarity, refining genetic counseling, and identifying targets for gene therapy, it’s important to note that genetic testing does not currently serve as a screening tool for IRDs [3]. Therefore, an accurate clinical examination remains pivotal for both the diagnosis of RP and its masquerade conditions.

Here, we present two cases resembling RP, emphasizing notable aspects from clinical signs and multimodal retinal imaging, and provide a comprehensive review of RP masquerades. By elucidating the distinct clinical features and key imaging findings of RP masquerades, we hope to aid the examiner in making an accurate clinical diagnosis. 


**Case 1:**


An 81-year-old female was referred to the IRD clinic for bilateral, slowly progressive dimming vision over 8 years. She had no known family history of IRD and denied nyctalopia in her youth. She did not report any syndromic features of RP, including hearing loss. Her visual acuity measured 20/400 in the right eye (attributable to a pre-existing large macular hole) and 20/80 in the left eye. Pupils were briskly reactive without an afferent pupillary defect. The slit lamp examination was unremarkable except for a poor tear film bilaterally. A dilated fundus exam revealed bilateral, diffusely scattered yellow-white punctate lesions along the vascular arcades, extending to the periphery with pigmentary changes (Figure 1A,B). Fundus autofluorescence showed bilateral bull’s eye maculopathy (Figure 1C,D). Spectral domain ocular coherence tomography (SD-OCT) showed the pre-existing large macular hole in the right eye, severe bilateral attenuation of the outer nuclear layer (ONL), and a ratty appearance of the ellipsoid zone (EZ), and retinal pigment epithelium (RPE) with abnormal hyperreflective signals (Figure 2). An electroretinogram (ERG) showed diminished scotopic and photopic responses (Figure 3). While providing further history to exclude non-IRD conditions, the patient disclosed a history of bariatric surgery nearly 30 years prior, followed by an intestinal obstruction requiring a colectomy 1 year ago. She had been on iron and vitamin B12 supplements since her bariatric surgery but had not been on a vitamin A replacement. The patient was ultimately found to have a severe vitamin A deficiency with a serum retinol level of 9.4 mcg/dL (normal values range from 20 to 60 mcg/dL) that caused xerophthalmia with retinopathy. She was immediately treated with intramuscular vitamin A supplementation (100,000 units/day for 3 days, then 50,000 units/day for 2 weeks and 100,000 units per day for 3 months). She was then maintained on 5000 to 10,000 units of vitamin A daily. Her visual acuity improved to 20/350 in the right eye and 20/40 in the left eye, and SD-OCT showed partial restoration of the EZ band (Figure 4), which suggests that adequate vitamin A treatment can reverse some of the vision loss and prevent further vision loss in vitamin A deficiency (VAD). 


**Case 2:**


A 71-year-old female presented with bilateral peripheral visual field restriction and nyctalopia for several years. Her past medical history was notable for lupus, for which she was treated with hydroxychloroquine (Plaquenil^®^) 200 mg twice daily for twenty years with a cumulative dose of 5216 mg. Her visual acuity was 20/20 in the right eye and 20/20 in the left eye. Pupils were briskly reactive without an afferent pupillary defect. The slit lamp examination was unremarkable. The dilated fundus exam showed optic disc pallor, arteriolar narrowing, peripheral RPE clumping, and retinal atrophy in both eyes (Figure 4). Goldman visual fields showed large pericentral scotomas bilaterally (Figure 5), and SD-OCT showed characteristic loss of the parafoveal EZ and ONL (Figure 6). The patient was diagnosed with advanced hydroxychloroquine retinopathy. Upon providing further history, the patient reported she received infrequent eye exams without focused hydroxychloroquine screening. After the diagnosis of hydroxychloroquine toxicity, her hydroxychloroquine was immediately discontinued.

## 2. Review of RP Masquerades

While the clinical symptoms and ocular findings of RP have been extensively characterized, the specific age of onset, degree of visual impairment, fundus findings, and rate of progression exhibit substantial variability due to genetic heterogeneity and other environmental influences [3]. The archetypal fundus appearance of RP includes a clinical triad of bone spicule intraretinal pigmentation, attenuation of retinal vessels, and waxy pallor of the optic nerve [3]. Fundus abnormalities are typically present bilaterally, with greater pigmentation in the midperiphery, corresponding to the region of highest rod cell concentration [5,6]. Bone spicule pigmentary changes within the retina represent the migration of pigment from disintegrated RPE cells into the interstitial spaces of the retina. Importantly, bone spicule pigmentary changes are neither specific nor sensitive for RP, serving instead as indicators of photoreceptor and/or RPE impairment resulting from a spectrum of retinal conditions and injuries. Furthermore, cystoid macular edema (CME) can arise in RP patients with a prevalence of 10–50% on optical coherence tomography (OCT); this can overlap with CME secondary to various retinal diseases [7]. In this context, we review various retinal diseases that emulate RP and discuss the salient distinguishing elements characterizing RP masquerade conditions (Table 1).

## 3. Metabolic: Vitamin A deficiency

Vitamin A, a fat-soluble nutrient, is absorbed in the small intestine and stored in the liver. It serves a crucial role in the visual phototransduction of rods and cones in the retina after being converted to retinol. Vitamin A also plays a pivotal role in ocular surface maintenance, and vitamin A deficiency (VAD) can result in retinopathy as well as conjunctival (Bitot’s spots) and corneal xerosis. 

VAD is common in the developing world due to malnutrition but rare in the developed world. When present, it is often linked to bariatric surgery or liver disease [8]. A cohort study reported that 69% of patients who undergo biliopancreatic diversion develop VAD within 4 years of surgery [9]. Furthermore, as in our case, signs and symptoms of VAD can occur years after bariatric surgery, as the liver can maintain considerable vitamin A stores. This may lead to underdiagnosis and delays in treatment. Thus, vitamin A supplementation is critical in patients who have undergone bariatric surgery or bowel resection, especially in patients with underlying liver disease. Due to the high prevalence of VAD with underlying medical conditions, it is crucial to obtain a focused medical history when suspecting a diagnosis.

The ocular manifestations of VAD can mirror those of RP due to the shared photoreceptor dysfunction that occurs in both entities. The earliest symptom is nyctalopia, but patients can also develop xanthopsia, constricted peripheral visual fields, and eventually decreased central vision. The clinical exam can reveal punctate retinal spots and a pigmentary retinopathy resembling RP, with bone corpuscle pigmentation in the midperiphery, areas of retinal atrophy, and arteriolar attenuation. ERG findings can be suggestive of rod–cone diseases [10]. 

A notable inherited form of vitamin A deficiency is abetalipoproteinemia, an autosomal recessive disorder marked by defective intestinal absorption of fat-soluble vitamins. In patients with abetalipoproteinemia, treatment with vitamin supplementation can reverse ERG abnormalities and improve dark adaptation [10]. 

Both acquired and inherited forms of VAD typically present bilaterally, but they can be less symmetric and exhibit faster progression than RP. Photoreceptor changes in SD-OCT, such as an irregular ratty appearance and abnormal hyperreflective signals along the ellipsoid zone (EZ) and RPE, observed in our patient (Figure 2), may present both in the center and peripheral retina from the onset of VAD. Wide-field fundus autofluorescence imaging is often helpful in facilitating early VAD detection. Because the complications of VAD have a reversible component through appropriate vitamin A supplementation, it is important for providers to be aware of this entity in the differential of RP. In contrast, vitamin A supplementation has been shown to have no effect on clinically relevant measures of visual function in patients with RP [11]. 

## 4. Drug-Induced

Retinal toxicity due to quinolines, including hydroxychloroquine (as described in Case 2), chloroquine, and quinine, can similarly yield an RP-like presentation. These drugs were historically employed for malaria treatment [12]. Long-term use of these medications can cause visual field constriction, inner retinal atrophy, vascular attenuation, and optic nerve pallor [13]. Furthermore, hydroxychloroquine toxicity carries the added complication of toxic maculopathy [14].

Hydroxychloroquine, now frequently employed in the treatment of rheumatologic conditions like lupus, Sjogren’s syndrome, and rheumatoid arthritis, can prompt a toxic retinopathy at elevated cumulative doses [12]. The incidence of hydroxychloroquine toxicity is 0.68% based on a 2010 study of about 4000 patients [15]. Despite the rarity of toxicity, routine screening is imperative, as retinal toxicity might remain asymptomatic during early stages, potentially culminating in severe and irreversible vision loss, even persisting after medication cessation. This is because hydroxychloroquine binds to melanin in the RPE, resulting in irreversible photoreceptor loss and RPE atrophy [14]. In advanced stages, patients may experience changes in their visual acuity, restricted peripheral visual fields, and nyctalopia [14]. Clinical examination can reveal bilateral macular RPE changes commonly referred to as the “bull’s eye” appearance, alongside optic disc pallor, vascular attenuation, and even peripheral bone spicule formation [14]. Given that our patient presented with an advanced stage of toxicity, she exhibited many of the fundus findings as well as OCT and visual field results. Notably, our patient had many risk factors for toxicity, including age greater than 60 years old, high cumulative dose (5216 mg), as well as a long duration of use (20 years). In fact, the most pivotal predictor of toxicity is the duration of use, for which cumulative dose can be a proxy [15]. Other recognized risk factors which were not present in our patient include underlying retinal disease and renal or liver disease, as hydroxychloroquine is metabolized in the liver by the cytochrome P450 enzyme system and the metabolites are excreted in the urine [15]. The ocular manifestations of advanced hydroxychloroquine toxicity can mimic RP due to the photoreceptor damage that occurs in both entities. Both entities can present with nyctalopia and peripheral visual field restriction. However, hydroxychloroquine toxicity will manifest in the setting of medication use. Because patients rarely notice visual symptoms in early stages, routine screening for hydroxychloroquine toxicity is essential. Recommended screening tests include automated visual field tests, SD-OCT imaging, multifocal ERG (mfERG), and fundus autofluorescence (FAF) [84] to detect subtle changes. SD-OCT findings include parafoveal EZ loss, inner retinal atrophy, and loss of the outer retinal layers. OCT is especially useful and is becoming an essential element of primary screening [13]. 

Phenothiazines, a class of antipsychotic medications including thioridazine and chlorpromazine, are associated with various ocular and dermatologic side effects. Prolonged high dose usage can induce a bilateral symmetric pigmentary retinopathy, predominantly linked to thioridazine and, to a lesser extent, chlorpromazine [17]. Pigmentary changes primarily affect the peripheral retina first and gradually encroach inwards, resulting in nyctalopia, loss of peripheral vision, and ultimately a central scotoma and blindness [18,19]. Keeping this condition on the differential for RP is important because symptoms may predate retinal changes, and visual acuity has been reported to improve after cessation of thioridazine [20]. However, it is worth noting that retinal pigmentary changes may persist and even advance despite drug discontinuation [21].

The risk of retinopathy in many of these drugs has been demonstrated to be dose dependent. For example, the maximum recommended daily dosage of thioridazine and chlorpromazine is 800 mg/day [17,85], and retinal toxicity with quinine is typically reported in patients who consume more than 4 g [86]. The maximum recommended daily dose of hydroxychloroquine is 5.0 mg/kg/day [84]. Consequently, despite the rare occurrence of toxicity related to these drugs, regular screening is advised to identify early changes even when patients might be symptom-free, aiming to avert irreversible vision loss. 

## 5. Chorioretinal Infections

TORCH infections, including toxoplasmosis, rubella, cytomegalovirus (CMV), herpes simplex virus (HSV), and infections with other organisms including varicella zoster and syphilis comprise a cluster of disorders transmitted during pregnancy. These infections can result in retinal pigmentary changes that bear resemblance to RP; however, unlike RP, these conditions cease progression following treatment. 

Toxoplasma gondii is an obligate intracellular parasite which can be transmitted to the fetus transplacentally, resulting in congenital toxoplasmosis, or acquired by ingesting contaminated food or water [38]. Ocular toxoplasmosis presents as unilateral white lesions with vitritis in the active phase and chorioretinal scarring in latent disease [39]. Secondary complications include fibrous bands, retinal detachments, optic neuropathy, and choroidal neovascular membranes [39]. Atypical manifestations of ocular toxoplasmosis that may mimic RP have also been documented [40]. Silveira et al. and Basta et al. have reported instances of ocular toxoplasmosis with unilateral pigmentary retinopathy akin to RP [41,42]. Similar to RP, fundus findings can display bone-spicule-shaped pigment deposits, optic disc pallor, arteriole constriction, and CME [42]. Furthermore, visual field constriction and nonrecordable mfERG readings can be seen [42]. However, in contrast to RP, these changes are generally unilateral and may only involve a portion of the fundus [40,42]. Serologic tests for anti-T. gondii IgG antibodies cannot provide a confirmatory diagnosis because of the large global population that is positive [40]. However, polymerase chain reaction (PCR) assays of ocular fluid samples, in conjunction with a thorough review of the patient’s history and genetic testing, can help exclude RP and confirm the diagnosis. 

Congenital rubella syndrome arises from vertical transmission of rubivirus during pregnancy. It is characterized by an array of systemic complications, with ocular involvment being the most prevalent (78%), followed by sensorineural deafness (66%), psychomotor retardation (62%), cardiac defects (58%), and mental retardation (42%) [43]. Ocular manifestations span microphthalmia, cataracts, glaucoma, and a non-progressive pigmentary retinopathy, occuring in 13.3% to 61% of cases [43]. This classic “salt and pepper” fundus appearance stems from a disruption of RPE embryogenesis [44] and can mirror the peripheral retinal changes seen in RP. However, patients with rubella retinopathy have minimal to no changes in their visual acuity and exhibit near normal ERG. Fundus autofluorescence shows stippled hypoautofluorescent and hyperautofluorescent changes, while OCT findings include EZ disruption, subretinal deposits, and RPE irregularities [45]. Systemic features and high titers of immunoglobulin G rubella can confirm the diagnosis. 

CMV retinitis is caused by cytomegalovirus, a double-stranded DNA virus that predominantly affects immunocompromised patients, with those with CD4 counts below 50 facing heightened susceptibility [46]. It can cause progressive full-thickness retinal inflammation, chiefly in the posterior pole along the vasculature, without vitritis. Over time, this inflammation can evolve into necrosis, pigment epitheliopathy, and optic atrophy [52], which must be distinguished from RP. Affected infants may present with a similar active retinitis. Although the disease is considered self-limiting in immunocompetent infants and systemic treatment can be detrimental, it’s worth noting that systemic therapy has demonstrated efficacy in active retinitis [47,48]. After the resolution of active retinitis, bilateral broad pigmentary changes may mimic RP. However, this typically does not further progress once the infection is adequately controlled. 

Herpes simplex virus (HSV) most commonly manifests as keratitis or keratouveitis but can also affect the retina. HSV1 is spread by direct contact, usually via saliva, and HSV2 is caused by sexual contact or transmitted from mother to child during birth [49]. Affected infants may present with bilateral pigmented chorioretinal scars that mimic RP [50].

Varicella Zoster virus (VZV), although not classified as a TORCH infection, is another herpes virus capable of affecting the retina. Affected infants can develop congenital varicella syndrome, featuring retinitis with chorioretinal scars and optic nerve hypoplasia, following a maternal varicella infection during the second trimester of pregnancy. Children diagnosed with congenital varicella syndrome should be examined by an ophthalmologist to exclude ocular abnormalities [51]. 

Syphilis is a sexually transmitted infection caused by the spirochete Treponema Pallidum. The natural course of syphilis spans four stages: primary, secondary, latent, and tertiary, and ocular involvement primarily occurs during the secondary stage [22]. Ocular syphilis can affect any ocular structure and manifests in a variety of ways, including uveitis, retinitis, papillitis, neuroretinitis, vasculitis, chorioretinitis, and panuveitis [23]. This diverse spectrum in presentation renders syphilis capable of mimicking RP. A 2021 case report describes a patient with nyctalopia and bilateral progressive vision loss for 3 years, with hand motion visual acuity in both eyes at presentation. Examination revealed anterior vitreous cells, arteriolar attenuation, diffuse RPE atrophy, and retinal pigmentary changes bilaterally, with extinguished ffERG readings in both eyes. Initially presumed to have RP, this patient was later diagnosed with syphilis, as confirmed by a positive FTA-Abs immunoglobulin G test [24]. In contrast to patients with RP, patients with ocular syphilis may present with a unilateral or asymmetric bilateral patchy chorioretinitis that can rapidly progress, culminating in chorioretinal scars in late stages [25]. Clinical diagnosis can be validated with positive specific treponemal tests. Timely recognition and appropriate antibiotic therapy can curtail vision loss in ocular syphilis patients [26]. The most common manifestation of congenital syphilis is bilateral interstitial keratitis. Pigmentary retinitis and secondary glaucoma can also occur as a result of congenital syphilitic keratouveitis [27]. 

Human immunodeficiency virus (HIV) is a virus that replicates in CD4 T-cells and is transmitted by exposure to bodily fluids. Roughly half of patients with HIV develop an ocular microvasculopathy characterized by cotton wool spots, microaneurysms, and intraretinal hemorrhages but little to no intraocular inflammation. It is usually asymptomatic and does not require additional ocular treatment other than systemic anti-HIV treatment. Beyond this, ocular involvement predominantly emerges in the context of opportunistic infections, with CMV retinitis being the most common [28], as discussed above.

Measles is a virus that can either be transmitted vertically during pregnancy or acquired. Infants with congenital measles present with systemic features including a rash, cardiomyopathy, and deafness, but they can also have ocular consequences including cataracts and retinal pigmentary changes. The retinal findings are described as scattered, granular black pigmentary changes which contrast with the typical bone corpuscles seen in RP. Unlike RP, there is no vascular attenuation or disc pallor in measles retinopathy. Furthermore, the patient’s visual function is usually intact and ERG findings are normal. 

Lyme disease is caused by Borrelia burgdorferi, a spirochete carried by ticks. Ocular manifestations vary and include intraocular inflammation, retinal vasculitis, and ischemic optic neuropathy [32,33,34]. Additionally, the results of serum antibody titers to B. burgdorferi may vary one from laboratory to another, further complicating the diagnosis [35]. The overall clinical presentation of Lyme disease may resemble RP. Karma et al. reported a patient with pigmentary retinopathy, optic neuropathy, and vitritis that mimicked RP but was associated with a positive Lyme PCR test from the vitreous and cerebrospinal fluid [36]. Timely diagnosis of ocular Lyme disease is important because signs and symptoms, including retinal vasculitis, can improve with tetracycline therapy [33].

Ocular tuberculosis (TB) is a rare extrapulmonary manifestation of the bacteria Mycobacterium tuberculum, an airborne infection that causes pathology primarily in immunocompromised patients [29]. Ocular involvement most commonly results from hematogenous dissemination to the uveal tract, favored by the choroid’s oxygen-rich environment, resembling that of the lungs. Ocular TB is considered a great imitator of many uveitic entities due to its wide spectrum of manifestations. It can cause granulomatous anterior, intermediate, or posterior uveitis [30]. Posterior involvement often features multifocal choroidal tubercles marked by discrete yellow lesions in the posterior pole that can be associated with an exudative retinal detachment [30]. A 2022 case report describes a patient who presented with vision loss and was found to have arteriolar attenuation, diffuse retinal hypopigmentation sparing the macula, pigment clumps in the midperiphery, severely constricted visual fields, and undetectable scotopic and photopic responses on an ffERG [31]. Subsequently, a positive Mantoux test of 18 mm induration confirmed TB. Traditional anti-tuberculosis treatment was administered with resolution of vascular leakage on FA [31]. In general, chest imaging, serology, and PPD skin testing in a patient with the appropriate clinical picture can provide support for the diagnosis, but a definitive diagnosis is only made when M. tuberculosis is found in a sample of intraocular fluid [30]. Treatment for ocular tuberculosis mirrors that for pulmonary tuberculosis, with the four-drug regimen of isoniazid, rifampicin, ethambutol, and pyrazinamide [30]. Because it is treatable, early recognition and diagnosis are imperative. 

Many infectious chorioretinitis conditions can present with wide spectrum of chorioretinal scars that may resemble RP, particularly after the resolution of the active phase. However, it’s important to note that these conditions typically do not exhibit further progression once the infection has been effectively treated. 

## 6. Noninfectious Inflammatory Diseases

There are several noninfectious inflammatory diseases which can resemble RP as well, such as chronic uveitis, pars planitis, rheumatologic conditions, and white dot syndromes.

Because young patients with RP can have fine colorless particles evenly distributed throughout the vitreous, RP can mimic a uveitic entity such as chronic uveitis or pars planitis. Pars planitis is a chronic, idiopathic intermediate uveitis that has been reported to mimic RP when bilateral and is associated with retinal pigmentary changes [87]. In contrast to the fine particles seen in RP, pars planitis typically presents with larger snowball-like aggregates that settle at the bottom of the vitreous cavity, along with peripheral retinal vasculitis, optic disc edema, and anterior segment inflammation [88]. CME is a common secondary complication. Generally, mild cases of pars planitis are not treated, but severe inflammation is usually managed with steroids or immunosuppressive agents to improve the prognosis [88].

Systemic rheumatologic conditions including sarcoidosis, lupus, and rheumatoid arthritis can also give rise to inflammatory conditions that mimics RP. The most common ocular manifestation of sarcoidosis is a chronic, bilateral, granulomatous anterior uveitis [63], though sarcoidosis can also trigger a posterior uveitis [64]. Posterior involvement can encompass choroidal granulomas or multifocal choroiditis with midperipheral periplephlebitis, manifesting as scattered yellow perivenular retinal exudates that have been described as “candle-wax drippings” [65]. Lupus retinopathy results from immune-complex-mediated vascular injury [66] often presenting as cotton wool spots, microaneurysms, arteriolar narrowing, and retinal exudates [67]. Phlebitis and arteritis can also occur, resulting in vessel occlusions. Lastly, rheumatoid arthritis more frequently induces anterior segment manifestations like dry eyes, episcleritis, scleritis, and peripheral ulcerative keratitis, but it may also cause a retinal vasculitis [68]. Prompt recognition is paramount in these rheumatologic conditions, as ocular involvement may be the first manifestation of the disease, and the ocular consequences of these conditions are treatable. In cases of severe intraocular inflammation, steroids or immunosuppressive agents may be used.

Other noninfectious, inflammatory conditions that mimic RP include some of the white dot syndromes, namely acute zonal occult outer retinopathy (AZOOR), birdshot chorioretinitis, and serpiginous choroidopathy. AZOOR, a rare retinal disease, has a predilection for young myopic females with a history of autoimmune diseases [56,57]. The disease often presents unilaterally but can evolve to bilateral involvement over time, characterized by symptoms such as photopsia and acute loss of one or more zones of the visual field. These changes stabilize after 6 months and seldom advance further [57,58]. In contrast to RP, visual acuity remains minimally affected and a fundus exam can be normal at presentation [56,57,58]. However, fundus abnormalities are noted in advanced cases, including RPE atrophy, particularly in the peripapillary region, as well as arteriolar narrowing and pigment clumping, which can mimic RP [56,59]. In all stages of the disease, OCT can reveal EZ loss and thinning of the outer nuclear layer and RPE, while ERG can expose photoreceptor and RPE dysfunction [59,60,61]. Fundus autofluorescence serves as a useful tool for identifying and monitoring affected retinal areas, particularly when the fundus exam is normal [59,61,62]. As of now, there are no established therapeutic modalities for AZOOR. Notably, AZOOR differs from RP in that it results in focal zones of photoreceptor loss with a demarcating line of progression at the level of the outer retina, rather than the diffuse midperipheral loss seen in RP. 

Birdshot chorioretinitis is a rare condition that causes bilateral posterior inflammation with a distinctive presentation. In contrast to RP, the average age of onset is 50 years old, and there is a strong association with HLA-A29 positivity [53]. Symptoms at presentation include photopsia, nyctalopia, dyschromatopsia, and floaters [37]. A fundus exam reveals bilateral, symmetric, creamy white teardrop-shaped lesions throughout the posterior pole and midperiphery, with a mild vitritis and retinal vascular leakage seen on FA. Unfortunately, this disorder follows a relapsing and remitting course marked by recurring inflammatory episodes. Because inflammation of the retina is often followed by scarring, the retinal lesions in birdshot chorioretinitis can undergo pigmentary changes that have been reported to resemble RP [54]. In terms of prognosis, central visual acuity is preserved until advanced disease, at which point refractory CME, macular scarring, and choroidal neovascular membranes may contribute to vision loss [53]. Due to the progressive nature of the disease, steroid-sparing immunomodulators constitute the mainstay of therapy [53].

Serpiginous choroidopathy is a rare condition that is characterized by recurrent inflammation of the choroid, RPE, and choriocapillaris [55]. It typically presents with vision loss, metamorphopsia, or scotoma and is typically bilateral but asymmetric [89]. Fundus exam reveals a geographic pattern of choroiditis at the peripapillary region that intermittently extends centrifugally and may eventually involve the fovea [55,89]. Recurrences are common, and new lesions are often contiguous with the old ones. Prior episodes are evidenced by areas of RPE atrophy and pigment clumping, which can resemble changes seen in RP. Treatment is typically with steroids and immunosuppressive agents [55,89]. Early recognition and treatment are important because macular involvement occurs in up to 88% of patients with serpiginous choroiditis who are untreated [90].

## 7. Autoimmune and Paraneoplastic Retinopathies

Autoimmune retinopathy (AIR) comprises a spectrum of diseases wherein the immune system mistakenly targets and damages retinal antigens, resulting in retinal degeneration. A diagnosis of AIR can be established with elevated serum levels of antiretinal antibodies, such as anti-recoverin or anti-enolase, in conjunction with electrophysiological evidence of retinal damage [69,70,91]. 

Three forms of AIR have been identified, including cancer-associated retinopathy (CAR), melanoma-associated retinopathy (MAR), and non-neoplastic autoimmune retinopathy (npAIR). CAR has a predilection for women and develops after the age of 45, with an average age of 65 [71]. CAR has been described in a variety of cancer types, including small cell lung carcinoma [92], mixed Müllerian tumor [93], endometrial carcinoma [94], and uterine sarcoma [72]. MAR is more common in men than women and can present years after the melanoma has been diagnosed, often during the metastatic stage [71]. npAIR represents the most common form of AIR and presents similarly to CAR but without any associated malignancy. It typically affects younger patients with a personal or family history of autoimmune disease [75].

In general, patients with AIR present with rapidly progressive bilateral vision loss and photopsia over a span of weeks to months [69,73]. The presentation may be asymmetric between the eyes [69,73]. Unlike RP, most patients do not have a longstanding history of night blindness or a family history of retinal degeneration. The ocular exam can be unremarkable or demonstrate signs of panretinal degeneration without pigment deposits, vessel attenuation, optic nerve pallor, or RPE mottling or atrophy [69,70]. There is typically no intraocular inflammation. ERG findings include central or global cone abnormalities [73], severe rod and cone dysfunction in CAR, and negative waveforms in MAR [70]. OCT may show minimal findings in early stages of the disease or more pronounced changes in the inner retina as well as loss of the outer nuclear layer in advanced cases [85,86,87]. FAF imaging has demonstrated areas of abnormal hyper- or hypoautofluorescence [70,74]. The diagnosis is confirmed if the patient tests positive for antiretinal antibodies, such as anti-recoverin or anti-enolase [69,70,74,77]. Currently, the most effective long-term treatment for AIR is immunosuppression [69], in addition to managing the underlying cancer if present.

## 8. Miscellaneous

Unilateral RP is rare but has been reported [95,96,97,98]. The differential for unilateral pigmentary changes and visual field loss includes prior blunt trauma, retinal detachment, and retained occult intraocular foreign bodies. Chronic retinal detachment can share similarities with RP due to RPE atrophy and pigment migration. Prior trauma can result in chorioretinal scarring and poor vision, which similarly may mimic RP. In fact, it has been reported that 20% of patients with retinal abnormalities stemming from a history of blunt ocular trauma can develop RPE sequelae [78]. Lastly, retained occult intraocular foreign bodies may induce changes reminiscent of the pigmentary changes seen in RP. They can cause damage via direct entry into the eye and subsequent ricocheting inside the eye, resulting in a pigmentary retinopathy caused by metabolic changes like oxidative stress [79]. Specifically, iron or copper foreign bodies can cause siderosis bulbi or chalcosis [80,81], respectively. Affected patients may present with intraocular inflammation and develop progressive pigmentary degeneration. Recognition of these conditions is important because ERG changes may be partially reversible in some cases [82,83].

## 9. Conclusions

In this review, we present two cases of pseudo-retinitis pigmentosa and review the clinical characteristics of RP and its associated masquerades. Early recognition and diagnosis of these often-treatable masquerades can mitigate or even reverse vision loss. By raising the clinician’s index of suspicion towards these RP mimics, this information can help the clinician make an accurate and timely diagnosis, ultimately improving patient outcomes. 

## Figures and Tables

**Figure 1 jcm-12-05620-f001:**
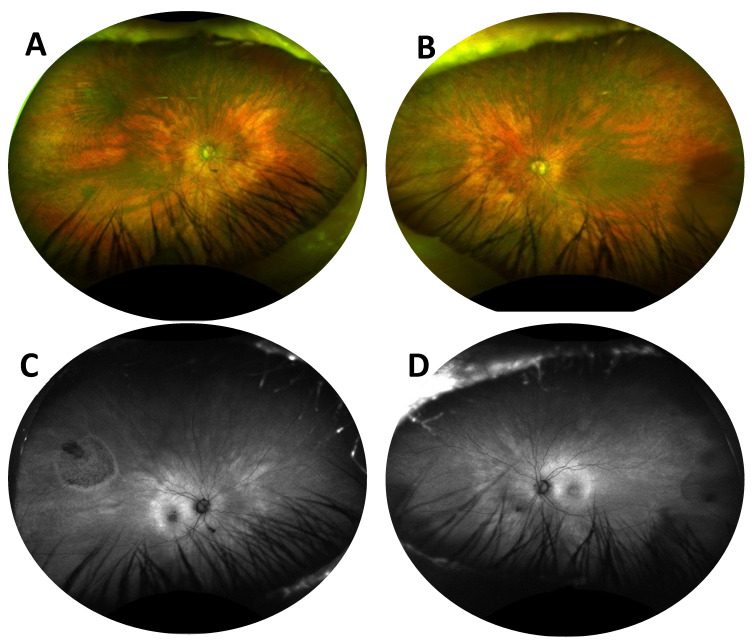
Fundus photos of the right (**A**) and left (**B**) eyes showing bilateral, diffusely scattered yellow-white punctate lesions along the vascular arcades extending to the periphery with pigmentary changes, and fundus autofluorescence of the right (**C**) and left (**D**) eyes showing bilateral bull’s eye maculopathy.

**Figure 2 jcm-12-05620-f002:**
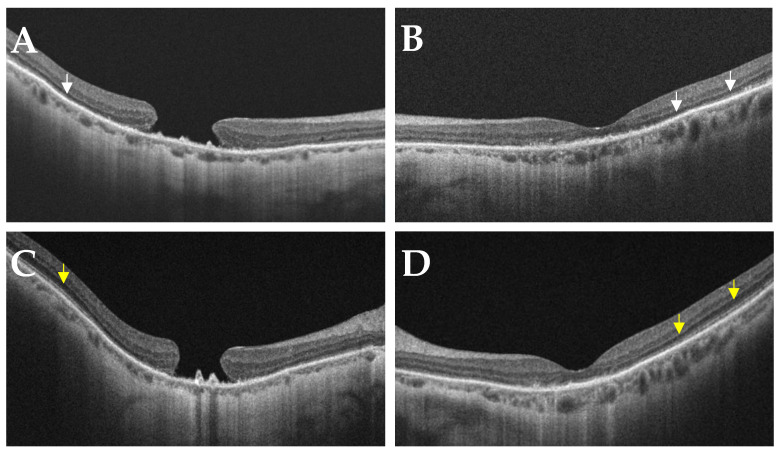
Spectral domain optical coherence tomography (SD-OCT) of the right (**A**) and left (**B**) eyes before treatment showing the pre-existing large macular hole in the right eye, severe bilateral attenuation of the outer nuclear layer (ONL), and a ratty appearance with abnormal hyperreflective signals along the ellipsoid zone (EZ) (white arrow heads) and retinal pigment epithelium (RPE). SD-OCT of the right (**C**) and left (**D**) eyes after treatment showing partial restoration of the EZ band (yellow arrow heads).

**Figure 3 jcm-12-05620-f003:**
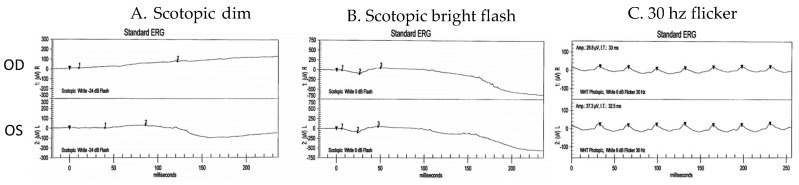
ERG of the right (top panels) and left (bottom panels) eyes showing diminished scotopic and photopic responses. (**A**) Scotopic dim flash, (**B**) scotopic bright flash, and (**C**) 30 Hz flicker.

**Figure 4 jcm-12-05620-f004:**
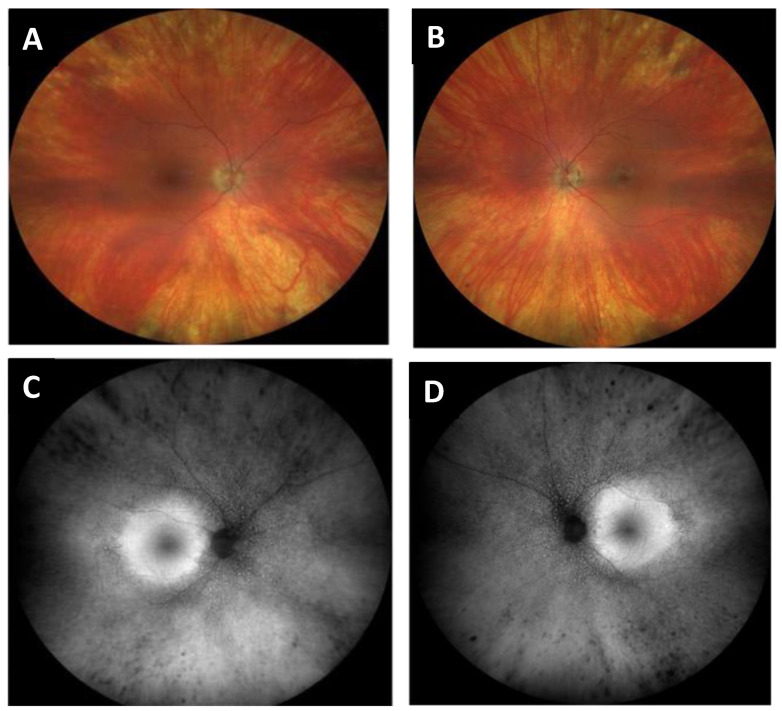
Fundus photos of the right (**A**) and left (**B**) eyes showing optic disc pallor, arteriolar narrowing, peripheral RPE clumping and retinal atrophy in both eyes. Fundus autofluorescence of the right (**C**) and left eyes showing bilateral bull’s eye maculopathy (**D**).

**Figure 5 jcm-12-05620-f005:**
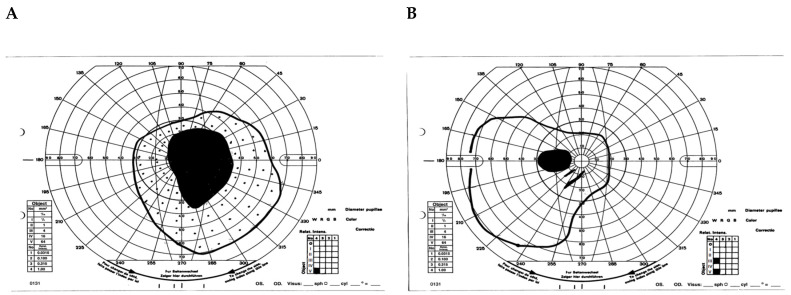
Goldman visual fields demonstrating large pericentral scotomas, more prominent in the right (**A**) than left (**B**) eye.

**Figure 6 jcm-12-05620-f006:**
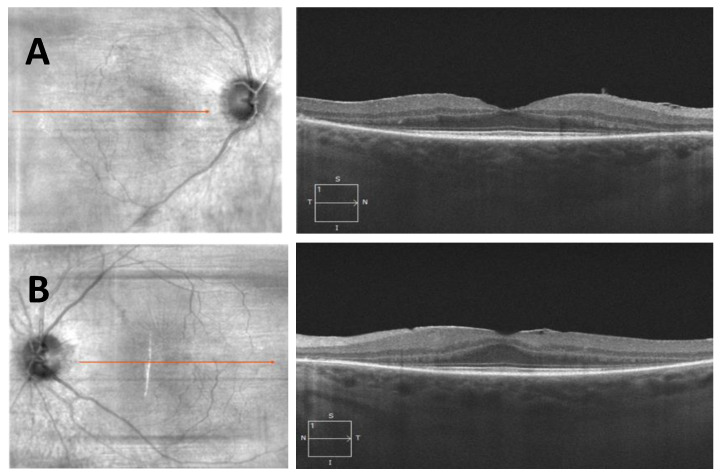
Spectral domain optical coherence tomography of the right (**A**) and left (**B**) eyes showing characteristic loss of the parafoveal EZ and outer retinal layers.

**Table 1 jcm-12-05620-t001:** Summary of Several Retinal Diseases that can Mimic RP, along with their Distinguishing Features.

		Distinguishing Features
Metabolic	Vitamin A deficiency [8,9,10,11]	Bilateral, often asymmetric with relatively rapid progression Conjunctival and corneal xerosis History of malnutrition, bariatric surgery, or liver disease Positive response to Vit A supplementation
Drug-Induced	Quinolines (quinine, chloroquine, hydroxychloroquine) [12,13,14,15,16]	History of long-term medication use Bull’s eye maculopathy Renal or liver disease No family history of night blindness or retinal degeneration
	Phenothiazines (thioridazine, chlorpromazine) [17,18,19,20,21]	
	Didanosine	
Chorioretinal infections	Bacterial: Treponema pallidum [22,23,24,25,26,27,28], Tuberculosis [29,30,31], Borrelia burgdorferi [32,33,34,35,36,37]	Can be transmitted via pregnancy Can be associated with systemic features Do not progress after treatment PCR assays of ocular fluid samples may be helpful No family history of night blindness or retinal degeneration
	Parasitic: Toxoplasma [38,39,40,41,42], Diffuse Unilateral Subacute Neuroretinitis (DUSN)	
	Viral (Rubella [43,44,45], CMV [46,47,48], HSV [49,50], VZV [51], HIV [28,52], Measles)	
Noninfectious Inflammatory diseases	Pars planitis	Young females with a history of autoimmune disease and characteristic retinal findings Can be relapsing and remitting or progressive Systemic workup may elucidate underlying rheumatologic conditions
	Birdshot and serpiginous chorioretinopathy [37,53,54,55]	
	Acute zonal occult outer retinopathy (AZOOR) [53,56,57,58,59,60,61,62]	
	Systemic sarcoidosis [63,64,65], lupus [66,67], and rheumatoid arthritis [68])	
Autoimmune and paraneoplastic retinopathies	Cancer-associated retinopathy (CAR) [69,70,71,72,73,74]	Rapidly progressive, asymmetric High serum levels of antiretinal antibodies, like anti-recoverin or anti-enolase Underlying malignancy No family history of night blindness or retinal degeneration
	Melanoma-associated retinopathy (MAR) [71]	
	Non-neoplastic autoimmune retinopathy [69,70,75,76,77]	
Miscellaneous	Trauma [78]	Unilateral History of prior trauma or intraocular foreign body No systemic features or family history of night blindness or retinal degeneration
	Chronic retinal detachment	
	Metallic foreign bodies (siderosis bulbi or chalcosis) [79,80,81,82,83]	
	Central serous chorioretinopathy (CSCR)	
	Laser scars	

## Data Availability

No new data were created or analyzed in this study. Data sharing is not applicable to this article.

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
