# Peer review of "Retinitis Pigmentosa Masquerades: Case Series and Review of the Literature"

_jcm, 2023, doi:10.3390/jcm12175620_

Round 1

Reviewer 1 Report

Dear Authors,

I wish to submit my review of the article "Retinitis Pigmentosa Masquerades: Case series and review of literature".

The article is well-written and exhaustively describes the RP masquerades. The authors should be commended for their work.

Nevertheless, I have some suggestions that could improve the manuscript.

The manuscript would greatly benefit from a table with key distinguishing factors of any RP masquerades described in the text.

In addition, the review was structured as a list of RP masquerades and their features. A paragraph or chart highlighting the main differences and the "flags" for differential diagnosis would be helpful for clinicians approaching the differential diagnosis. 

Author Response

Dear reviewer, 

We appreciate the your valuable comments to improve our manuscript. We revised our table adding key pertinent findings of differential diagnostic criteria.

While the clinical manifestations of each diagnosis share overlap and lack singular exclusive points, we hope that the revised table can provide initial guidance during the diagnostic differentiations. 

Sincerely,

Sun Young Lee 

Reviewer 2 Report

In the manuscript Abinaya Thenappan et al. presented two pseudo retinitis pigmentosa (RP) cases (Vitamin A deficiency and hydroxychloroquine toxicity) and provided a comprehensive review of clinical characteristics of RP and RP masquerades. The strength of the manuscript is the detailed review of clinical characteristics of RP masquerades. In the opinion of this reviewer, the only point of weakness of this manuscript is the cases presentation section, indeed it will be useful for the readers to present another case of a younger patient whit an infectious disease where the differential diagnosis could be more challenging.  

I read this manuscript with great interest. The article is well written and it may play an important role in the diagnosis and management of patients with RP masquerade conditions. 

Below my recommendation

“Table 1. Summary of several retinal diseases that can mimic retinitis pigmentosa.”: please add references or another table in supplementary material with a reference for each disease.

Author Response

Dear reviewer, 

We appreciate the your valuable comments to improve our manuscript. We revised our table adding references. 

While the suggestion for adding a case of infectious diseases in youger patient could be valuable we decided not to add an additional case at this stage. Instead, we added additional summary of pertinent findings of each differentials in our table. While the clinical manifestations of each diagnosis exhibit some degree of overlap and lack singular exclusive points, we hope that the revised table will be helpful to provide an initial guidance in diagnostic determination. 

Sincerely,

Sun Young Lee